# Transcriptome Analysis Reveals Genes and Pathways Associated with Salt Tolerance during Seed Germination in *Suaeda liaotungensis*

**DOI:** 10.3390/ijms232012229

**Published:** 2022-10-13

**Authors:** Jieqiong Song, Xinjie Liu, Xinxin Li, Hongfei Wang, Ruowen Chu, Fangfang Qu, Sixue Zhang, Qiuli Li

**Affiliations:** Key Laboratory of Plant Biotechnology of Liaoning Province, School of Life Sciences, Liaoning Normal University, Dalian 116081, China

**Keywords:** salt stress, halophyte, *Suaeda liaotungensis*, seed germination, transcriptomic

## Abstract

Seed germination is susceptible to external environmental factors, especially salt stress. *Suaeda liaotungensis* is a halophyte with strong salt tolerance, and the germination rate of brown seeds under 1000 mM NaCl treatment still reached 28.9%. To explore the mechanism of salt stress response during brown seed germination in *Suaeda liaotungensis*, we conducted transcriptomic analysis on the dry seeds (SlD), germinated seeds under the control condition (SlG_C), and salt treatment (SlG_N). Transcriptome analysis revealed that 13314 and 755 differentially expressed genes (DEGs) from SlD vs. SlG_C and SlG_C vs. SlG_N were detected, respectively. Most DEGs were enriched in pathways related to transcription regulation and hormone signal transduction, ROS metabolism, cell wall organization or biogenesis, and carbohydrate metabolic process in two contrasting groups. Compared with the control condition, POD and CAT activity, H_2_O_2_, soluble sugar, and proline contents were increased during germinated seeds under salt stress. Furthermore, functional analysis demonstrated that overexpression of *SlNAC2* significantly enhanced salt tolerance during the germination stage in *Arabidopsis*. These results not only revealed the tolerant mechanism of brown seed germination in response to salinity stress but also promoted the exploration and application of salt-tolerant gene resources of *Suaeda liaotungensis*.

## 1. Introduction

Soil salinization is an increasingly agricultural problem worldwide that affects the yield and quality of crops. Salt stress affects more than 6% of the world’s land area, as well as 20% of cultivated land area [1]. The accumulation of salt in cultivated land mainly comes from irrigation water and seawater containing a high concentration, usually around 400–600 mM, of sodium chloride (NaCl), which is caused by a variety of variables, including inappropriate irrigation methods, salinization irrigation, and seasonal impacts [1]. Each year, 1.5 million hectares of land are unproductive because of excess salinity in soil, and the serious salinization of more than 50% of all arable lands may be caused until 2050 [2]. Therefore, it is critical to cultivate salt-tolerant crops and increase crop yield. The identification of key salt-tolerant genes and application of molecular breeding are highly effective strategies to cultivate salt-tolerant crops, thereby enhancing the ability to adapt to the saline environment.

Seed germination is the essential stage of seed plant ontogeny. It is a complicated process of plant physiology, biochemistry, and material metabolism, which is regulated by plant growth substances and influenced by external environmental factors. For good seed resources, rapid and uniform germination is required [3]. The seed germination process consists of three phases, including rapid imbibition phase, plateau phase, and radicle protrusion (RAP) [4]. The metabolic activation, vacuolization, and reserve mobilization of endosperm cells are the earliest indicators of germination, which are followed by embryo cell growth, testa rupture, endosperm rupture, and ultimately radicle emergence [5,6]. Phytohormones are vital to seed germination. Different hormones influence the seed germination process by regulating the equilibrium between abscisic acid (ABA) and gibberellins (GA) at the signaling levels [7,8]. Salt stress inhibits seed germination due to osmotic stress, oxidative stress, and ion toxicity, which mainly shows the lower germination rate and a longer germination period [9]. Among different stages of seed germination, the plateau phase is more vulnerable to salt stress than the imbibition phase, because the plateau phase provides resources and energy for seedling growth [3]. Moreover, excessive reactive oxygen species (ROS) are produced by salt stress, which triggers oxidative damage of cell constituents, including membrane lipids, proteins, and nucleic acids, thereby causing oxidative stress [1]. Meanwhile, salt stress disturbs the dynamic equilibrium between ROS formation and elimination by altering the activity of antioxidant enzymes such as superoxide dismutase (SOD), catalase (CAT), and peroxidase (POD) [10]. Many complex networks, such as those relevant to gene expression and controlled by transcription factors, hormone signaling, and secondary metabolism, are linked to salt stress during seed germination, according to transcriptome analysis [9,11].

*Suaeda liaotungensis* (Kitag), a succulent halophyte of *Chenopodiaceae* family, is found along China’s coasts. *S. liaotungensis* is an annual plant that lacks salt glands and salt bladders in leave, yet it has a strong resistance to salt and alkali stresses and thrives in saline-alkali environments. Additionally, *Suaeda* plants possess the ability to absorb heavy metals to improve heavy metals-contaminated soil, thereby reflecting its ecological value [12]. Preliminary studies of our lab find that the *S. liaotungensis* seeds are dimorphic, with brown and black seeds, and substantial variances in germination, dormancy, and salt tolerance, and brown seeds have a better salt tolerance than black seeds during germination. Most of the species from *Chenopodiaceae* have relatively stable chromosome organization with the basic number of 9. The diploid *Suaeda salsa* has nine pairs of chromosomes (2 *n* = 18). As the genomic data of *S. liaotungensis* have not been published, the studies on salt tolerance mainly focus on the salt-tolerant genes expression [13,14]. However, the critical salt-tolerance genes, as well as the underlying molecular pathways during seed germination in *Suaeda liaotungensis*, are unknown as a potential gene resource for salt stress resistance.

In this study, we selected dry seeds (SlD), germinated seeds under distilled water condition (SlG_C), and salinity stress treatment (SlG_N) from brown seeds with higher salt tolerance of *Suaeda liaotungensis* to study the salt-tolerant mechanism during germination. The transcriptome profiles from two comparison groups, namely SlD vs. SlG_C and SlG_C vs. SlG_N, were analyzed and compared using RNA sequencing (RNA-seq). Meanwhile, the changes in antioxidant enzyme (SOD, CAT, and POD) activities, H_2_O_2_, soluble sugar, and proline contents were detected in SlD, SlG_C, and SlG_N. Moreover, salinity tolerance of *SlNAC2* was also demonstrated during the germination stage in *Arabidopsis thaliana*. Our findings provide light on the molecular mechanisms underlying salt tolerance during brown seed germination in *Suaeda liaotungensis*. Some candidate genes would be studied further and used to breed salt-tolerant crops, thereby improving soil salinization.

## 2. Results

### 2.1. Salinity Tolerance of S. liaotungensis during Brown Seed Germination

To explore the suitable salt tolerance concentration during germination, germination experiments were performed under distilled water (the control) and different concentrations of NaCl treatment. With the further increase in salt concentration, the germination rate gradually decreased. The germination rate reached 90.8% under 500 mM NaCl treatment (Figure 1A). Under 600 mM NaCl condition, the germination rate began to show significant differences compared to the control condition (Figure 1A). Under the 1000 mM NaCl treatment, the germination rate was still 28.9% (Figure 1A). Therefore, 500 mM NaCl was selected as the suitable salt tolerance concentration during the germination stage. To collect samples for transcriptome sequencing, imbibition curves of brown seeds were performed under the control and 500 mM NaCl treatment. Under the control, brown seeds imbibed water rapidly, and seed mass increased by 44.1% ± 1.5% after 2 h, 76.3% ± 1.6% after 6 h, and 154.9% ± 16.4% after 24 h (Figure 1B). By contrast, under the salt stress, brown seeds imbibed water slowly, and seed mass increased by 34.3% ± 2.8% after 2 h, 77.5% ± 5.3% after 12 h, and 180.8% ± 7.6% after 48 h (Figure 1B). Therefore, the times of concentrated germination under the control and NaCl treatment were 24 h and 48 h, respectively (Figure 1B). Germinated seeds with the same traits were collected for further experiment and transcriptome analysis (Figure 1C–E).

### 2.2. Determination of Physiological Indexes under Salt Stress

Salt stress produces excessive ROS, which leads to oxidative damage during seed germination. However, the antioxidant enzymes such as SOD, CAT, and POD eliminate excessive ROS, thereby maintaining the dynamic equilibrium of ROS. Compared with SlD, there was a significant decline in POD activity and increase in CAT activity, whereas SOD activity had a non-significant change in SlG_C (Figure 2A–C). Under salt stress, POD and CAT activities and H_2_O_2_ content were increased compared with the distilled water during seed germination (Figure 2B–D). Compared with SlD, soluble sugar content and proline content in SlG_C were both significantly decreased, but these two indexes were both significantly increased during seed germination under salt stress compared with distilled water (Figure 2E,F).

### 2.3. Sequencing, De Novo Transcriptome Assembly and Annotation

To explain the salt-tolerant mechanism of *S. liaotungensis* during brown seed germination, the dry seeds, germinated seeds from distilled water, and NaCl solution were collected, and transcriptome sequencing was performed. Using an Illumina Novaseq 6000 platform, nine cDNA libraries (three replicates per treatment) were constructed. Transcriptome sequencing generated approximately 41.41–53.04 million raw reads for nine libraries (Figure 3A). After filtering and removing low-quality reads, 42.62, 40.92, 42.80, 48.32, 48.95, 52.47, 47.39, 45.47, and 47.86 million clean reads were produced for each experimental sample (Figure 3A). In total, 61 Gb clean data were obtained, with each sample yielding up to 6.01 Gb clean data, and Q30 percentage was more than 93.06% (Figure 3A). Using the Trinity approach, the clean reads from nine libraries were assembled and merged into 90,665 unigenes, and N50 average length was 1517 bp (Appendix A). The comparison rate of mapping results for each sample ranged from 79.17% to 81.36% (Appendix A). Of all assembled unigenes, 200–500 bp of unigenes accounted for 58%, 500–1000 bp accounted for 20%, and 9354 (10%) unigenes were over 2000 bp in length (Appendix A). These results indicate that the quality of assembled unigenes could be used to perform follow-up analysis.

Owing to the lack of reference genome for *S. liaotungensis*, all unigenes obtained by transcriptome sequencing were matched with six databases, including NR, COG, GO, KEGG, Swiss-Prot, and Pfam database, to obtain annotation information (Appendix A). Among 90,665 unigenes, a total of 89,122 (98.3%) unigenes were matched with the six public databases. Of all annotated unigenes, 26,951 (30.2%) unigenes annotated to GO database, 18,279 (20.5%) unigenes annotated to KEGG database, 32,016 (35.9%) unigenes annotated to COG database, 36,496 (41.0%) unigenes annotated to the NR database, 26,170 (29.4%) unigenes annotated to Swiss-Prot database, and 26,748 (30.0%) unigenes annotated to Pfam database (Appendix A).

### 2.4. Identification of Differentially Expressed Genes (DEGs)

Principal component analysis (PCA) revealed the differences among SlD, SlG_C, and SlG_N. As shown in Figure 3B, PC1 and PC2 accounted for 42.23% and 12.18% of the variance, respectively. There were significant differences between SlD and the other two groups. However, there was a slight difference between SlG_C and SlG_N groups as a result of similar germination rate and phenotype (Figure 3B). We screened the DEGs based on the|log_2_ fold-change| ≥ 1 and *p*-value < 0.05. To determine which DEGs were altered at two different germination stages, the transcriptome of SlD with that of SlG_C was compared. To determine which DEGs were induced by salt stress during germination, the transcriptome of comparison analysis between SlG_C and SlG_N was performed. In the SlD vs. SlG_C group, 13314 DEGs (7366 upregulated and 5948 downregulated) were identified, while 755 DEGs (376 upregulated and 379 downregulated) were identified in the SlG_C vs. SlG_N group (Figure 3C). These findings showed that from dry seed to the germination stage, the great majority of DEGs were altered, but only 755 DEGs are regulated by salt stress during the germination stage.

### 2.5. GO and KEGG Enrichment Analysis of DEGs

To gain more information concerning seed germination under salt stress, GO functional annotation was performed. All DEGs were divided into three categories based on their GO annotations: biological process, cellular component, and molecular function. In the SlD vs. SlG_C group, 13314 DEGs were annotated to 52 GO terms consisting of 23 biological processes, 14 cellular components, and 15 molecular functions; however, 755 DEGs were annotated to 41 GO terms consisting of 17 biological processes, 12 cellular components, and 12 molecular functions in the SlG_C vs. SlG_N group (Appendix A). In addition, GO enrichment analysis showed that 116 and 75 GO terms were considerably enriched in SlD vs. SlG_C and SlG_C vs. SlG_N group, respectively, with the findings of significantly enriched GO terms (*P*_adj_-value < 0.05) presented in Appendix A. Lipid metabolic process (GO: 0006629), metal ion transport (GO: 0030001), hormone-mediated signaling pathway (GO: 0009755), regulation of nucleic acid-templated transcription (GO: 1903506), regulation of transcription, DNA-templated (GO: 0006355), and regulation of RNA biosynthetic process (GO: 2001141) were specifically enriched in the SlD vs. SlG_C group; however, response to oxidative stress (GO: 0006979) and catabolic process (GO: 0009056) were both specifically enriched in the SlG_C vs. SlG_N group (Figure 4). Three biological processes were found to be enriched in SlD vs. SlG_C and SlG_C vs. SlG_N, including carbohydrate metabolic process (GO: 0005975), cell wall organization or biogenesis (GO: 0071554), and reactive oxygen species metabolic process (GO: 0072593) (Figure 4). The DEGs related to the three processes should be given more attention in further research.

To further identify important pathways related to salt stress during germination, KEGG enrichment analysis was performed. In the SlD vs. SlG_C group, plant hormone signal transduction, phenylpropanoid biosynthesis, pentose and glucuronate interconversions, MAPK signaling pathway-plant, and glycerolipid metabolism were the top five enrichment pathways (Appendix A). In the SlG_C vs. SlG_N group, pentose and glucuronate interconversions, phenylpropanoid biosynthesis, limonene and pinene degradation, arginine and proline metabolism, and ascorbate and aldarate metabolism were the top five enrichment pathways (Appendix A). Among these enrichment pathways, most DEGs were predominantly enriched in pathways including phenylpropanoid biosynthesis, plant hormone signal transduction, glycolysis/gluconeogenesis, plant-pathogen interaction, and amino sugar and nucleotide sugar metabolism in the SlD vs. SlG_C group (Appendix A). Nevertheless, most DEGs were predominantly enriched in pathways of the pentose and glucuronate interconversions, phenylpropanoid biosynthesis, amino sugar and nucleotide sugar metabolism, arginine and proline metabolism, and plant hormone signal transduction in the SlG_C vs. SlG_N group (Appendix A). Among the top ten enrichment pathways, pentose and glucuronate interconversions, phenylpropanoid biosynthesis, amino sugar and nucleotide sugar metabolism, and plant hormone signal transduction were common in two comparison groups. These results revealed that variations in expression levels of DEGs in these pathways played essential roles during brown seed germination under salt stress.

### 2.6. Identification of DEGs Associated with Transcription Factor and Hormone Signal Transduction

The transcriptome data identified 316 DEGs encoding transcription factors (TFs) in the SlD vs. SlG_C group, of which, there were 250 upregulated genes and 66 downregulated genes (Appendix A). These DEGs mapped to 26 different TF families that were important regulators during seed germination, such as MYB, AP2/ERF, bHLH, C2C2, B3, bZIP, WRKY, NAC, MADS, and GRAS families (Appendix A). Among them, 67.1% DEGs belonged to MYB, AP2/ERF, bHLH, C2C2, B3, bZIP, WRKY, and NAC families in the SlD vs. SlG_C group (Appendix A). However, only 14 TFs were differentially regulated, as listed in Figure 5A, and mapped to nine TF families in the SlG_C vs. SlG_N group, of which, 8 up-regulated genes and 6 down-regulated genes were responsive to salt stress during seed germination (Appendix A). Among them, 11 genes were expressed in both groups, while 3 genes (TRINITY_DN19891_c0_g1, TRINITY_DN26339_c0_g1, and TRINITY_DN8056_c0_g1) were changed significantly only in the SlG_C vs. SlG_N group (Figure 5A). Meanwhile, under salt stress, five DEGs showed higher expression, while one DEG showed lower expression in the SlG_N group than that in the other two groups (Figure 5A).

102 DEGs (79 up-regulated and 23 down-regulated) associated with the plant hormone signal transduction, including auxin, ABA, GA, CK, ET, BR, JA, and SA pathways, were identified in SlD vs. SlG_C (Appendix A). Among them, 72 DEGs were related to auxin, ABA, and BR pathways, which might actively participate in seed germination; additionally, 32 DEGs in auxin pathway and 7 DEGs in cytokinin pathway were up-regulated (Appendix A). However, only 9 DEGs in SlG_C vs. SlG_N were listed in Appendix A and responsive to salt stress during seed germination. Among them, one downregulated and two upregulated genes were found in the ABA pathway, while one upregulated and four downregulated genes were found in the BR pathway, and one downregulated gene was found in the CK pathway (Figure 5B and Appendix A). The expression levels of gene (TRINITY_DN12138_c0_g1) changed significantly in the SlG_C vs. SlG_N group, while there was no change in the SlD vs. SlG_C group (Appendix A).

Taken together, transcriptional regulation and plant hormone played vital roles in germination under salinity stress in *S. liaotungensis*.

### 2.7. DEGs Related to Metabolic Pathways

According to GO enrichment analysis, most DEGs were found to be enriched to three metabolism pathways, including reactive oxygen species metabolic process, cell wall organization or biogenesis, and carbohydrate metabolic process, in both groups. A total of 69 DEGs (60 up-regulated and 9 down-regulated) were found in the reactive oxygen species metabolic process, 107 DEGs (93 up-regulated and 14 down-regulated) were found in the cell wall organization or biogenesis, while 404 DEGs (324 up-regulated and 80 down-regulated) were found in the carbohydrate metabolic process in the SlD vs. SlG_C group (Appendix A). During seed germination under salt stress, 16 DEGs (6 up-regulated and 10 down-regulated) were found in the reactive oxygen species metabolic process, 21 DEGs (10 up-regulated and 11 down-regulated) were found in the cell wall organization or biogenesis, while 55 DEGs (24 up-regulated and 31 down-regulated) were found in the carbohydrate metabolic process in the SlG_C vs. SlG_N group (Appendix A).

In the reactive oxygen species metabolic process, 55 DEGs encoded peroxidase-like, superoxide dismutase, catalase-like, and ascorbate peroxidase, and most DEGs were significantly upregulated in the SlD vs. SlG_C group (Appendix A). Among 47 DEGs encoding peroxidase-like, 12 DEGs (3 upregulated and 9 downregulated) were induced by salt stress at germination stage (Figure 6A). In the cell wall organization or biogenesis, the majority of DEGs classified into some groups, such as xyloglucan endotransglucosylase/hydrolases (XTHs), expansin, cellulose synthase/cellulose synthase-like (CESA/CSL), pectin-modifying enzymes and polygalacturonase-like, were significantly upregulated in the SlD vs. SlG_C group (Appendix A). Among 16 DEGs in the SlG_C vs. SlG_N group, five genes encoding xyloglucan endotransglucosylase/hydrolases were repressed, whereas two genes encoding expansin were enhanced by salinity stress at germination stage (Figure 6B). Five genes encoding polygalacturonase were upregulated by salinity stress, of which, two genes (TRINITY_DN3973_c0_g1 and TRINITY_DN42752_c0_g1) showed no change in the SlD vs. SlG_C group (Figure 6B). Among these enzymes associated with pectin modification, two pectinesterase genes were upregulated, whereas two genes encoding pectinesterase inhibitors were repressed under salt stress at the germination stage. In the carbohydrate metabolic pathway, most of the DEGs related to processes, including the glyoxylate cycle, starch synthesis and catabolic process, sucrose synthesis, and glycolytic cycle, were upregulated in the SlD vs. SlG_C group (Appendix A). These processes provided the energy source for seed germination. A total of 92 DEGs (80 up-regulated and 12 down-regulated) encoding endochitinase, chitinase, glycosidase, and endoglucanase were found in the SlD vs. SlG_C group (Appendix A). Under salt stress, two DEGs encoding phosphoenolpyruvate carboxykinase (TRINITY_DN28830_c0_g1) and malate synthase (TRINITY_DN15255_c0_g3) in the glyoxylate cycle were repressed during germination (Figure 6C). Meanwhile, four DEGs encoding the core enzymes of the starch synthesis process, including ribulose bisphosphate carboxylase (TRINITY_DN4032_c0_g1), glucose-1-phosphate adenylyltransferase (TRINITY_DN480_c0_g1), fructose-bisphosphate aldolase (TRINITY_DN17700_c0_g1), and sedoheptulose-1,7-bisphosphatase (TRINITY_DN7313_c0_g1), were upregulated during germination (Figure 6C). In addition, eleven glycosidase genes, including glucosidase, arabinofuranosidase, galactosidase, and arabinosidase genes, and five genes encoding endochitinase and endoglucanase were significantly induced by salt stress during germination stage (Figure 6C and Appendix A). These results above indicate that these metabolic processes are closely associated with the salinity tolerance during seed germination in *S. liaotungensis*.

### 2.8. Verification of DEG by RT-qPCR

To validate the reliability of RNA-seq results, 15 DEGs were selected to perform the RT-qPCR experiment in dry seed and germinated seed under distilled water and 500 mM NaCl solution (Figure 7). Of these genes, 3 DEGs were up-regulated in the two groups. Seven DEGs were significantly increased in the SlD vs. SlG_C group, whereas these genes were significantly repressed in the SlG_C vs. SlG_N group. One DEG was down-regulated in the SlD vs. SlG_C group and up-regulated in the SlG_C vs. SlG_N group. Three DEGs were not significantly changed in the SlD vs. SlG_C group, but these genes were triggered by salt stress during germination stage. The expression trends of 15 DEGs in both groups were similar to the RNA-seq results except TRINITY_DN52120_c0_g1 (Figure 7A,B). The results obtained by RT-qPCR were highly consistent with the RNA-seq data (R^2^ = 0.80 from SlD vs. SlG_C and R^2^ = 0.94 from SlG_C vs. SlG_N) (Figure 7C,D). These results indicate that transcriptome data is beneficial to identify key genes for salt tolerance during seed germination in *S. liaotungensis*.

### 2.9. Functional Characterization of SlNAC2 in Seed Germination under Salt Stress in Arabidopsis

Transcriptome analysis revealed that *SlNAC2* (TRINITY_DN7999_c0_g1) was significantly up-regulated in two contrasting groups, and the expression level of *SlNAC2* was highest in the germinated seeds under salt stress. Overexpression of *SlNAC2* in *Arabidopsis* enhances the survival rate of seedlings under salt stress [15]. However, the salinity tolerance of *SlNAC2* at seed germination stage is unknown. Therefore, we used three independent *SlNAC2* overexpression lines in *Arabidopsis*, namely *SlNAC2-OX*, to detect the salinity tolerance during germination. RT-PCR analysis was performed to verify the expression levels of the three transgenic lines (Figure 8A). To further analyze the salt tolerance of *SlNAC2-OX* lines at germination stage, we performed the germination experiment under the control condition and 150 mM NaCl treatment. Without NaCl treatment, three *SlNAC2-OX* lines showed similar germination rates to that of wild-type and pBI121 plants (Figure 8B–D). Under 150 mM NaCl treatment, the germination percentages of three *SlNAC2-OX* lines were approximately 60% at 4 DAS, while the seeds from wild-type and pBI121 plants germinated only half that of these *SlNAC2-OX* lines (Figure 8E). The germination rates of wild-type and pBI121 plants were approximately 50%, while the seeds from three *SlNAC2-OX* transgenic lines already germinated over 80% at 7 DAS (Figure 8C–E). These results demonstrate that *Sl**NAC2* transcription factor acts as a positive regulator in salinity tolerance during seed germination in *S. liaotungensis*.

## 3. Discussion

Seed germination is vulnerable to environmental stresses, especially salt stress. The effects of salt stress on seed germination mainly show the decreasing germination rate and prolonged germination time, thereby causing osmotic stress, oxidative stress, and ion toxicity [9]. Traditional crops, such as corn, rice, and wheat, cannot grow in saline-alkali land, but the halophyte can grow well. *S. liaotungensis* is a halophyte with strong salt tolerance. *S. liaotungensis* has seed heteromorphism, including black and brown seeds, and this characteristic is an important reproductive strategy to adapt to saline environments. Previous investigations found that brown seeds have a higher salt tolerance than that of black seeds during germination. Thus, the salt-tolerant mechanism of brown seeds during germination was studied. In this study, the germination rate of brown seeds under 500 mM NaCl treatment was similar to that in the distilled water, while the germination rate under 1000 mM NaCl treatment still reached 28.9% (Figure 1A), suggesting that there are some special mechanisms to adapt to the salinity stress during germination in *S. liaotungensis*. However, the salt-tolerance mechanism of seed germination is still unknown. In this study, we combined physiological, molecular, and transcriptomic approaches to reveal the molecular mechanism in *S. liaotungensis* that responds to salt stress during germination. It is possible to provide good salt tolerance gene resources for breeding new salt-tolerant crops.

Salt stress leads to many responses in plants, including hormone signal transduction and transcription regulation, oxidation stress, cell wall modification, and energy and substrate metabolism [16,17]. In halophytes, many studies have reported some molecular mechanisms for salt tolerance [18,19]. Transcriptome analysis for salt tolerance in *Suaeda glauca* was conducted, and most DEGs were enriched to signal transduction, cell wall metabolism, transcription factors, ion transport, and energy metabolism [18,19]. Several processes, including hormone signaling pathway, amino acid catabolic process, lipid metabolism, photosynthesis, and cell wall organization or biogenesis, were shown to be enriched in salt-tolerant genotype during seed germination [20]. To further reveal the mechanism in response to salt stress during seed germination in *S. liaotungensis*, transcriptome analyses of SlD, SlG_C, and SlG_N were performed using the RNA-seq technology. During the germination stage, we found that 755 DEGs were induced by salt stress, of which, the majority of genes were involved in transcription regulation and hormone signal transduction, reactive oxygen species metabolism, cell wall organization or biogenesis, and carbohydrate metabolism (Figure 3C and Figure 4). According to expression at the transcriptome level, some genes possibly related to salt tolerance were found, which is conducive to revealing the salt-tolerant mechanism during seed germination in *S. liaotungensis* (Figure 9).

### 3.1. Plant Hormone Signal Transduction Involved in Seed Germination under Salt Stress

ABA and GA antagonistically regulate seed dormancy and germination, and the other hormones play roles in this process by controlling the ABA/GA balance at biogenic or signaling levels [8]. Most genes were related to ABA, auxin, BR, ethylene, and SA signaling pathway from dry seed to germinate stage in *S. liaotungensis*, which suggested that these hormones played important roles in the process (Appendix A). Under salinity stress, the effects of salt stress on germination could be alleviated by ABA [21]. Protein phosphatases type 2C (PP2Cs) act as negative regulators of ABA signaling [22]. Overexpressing *OsPP108A*, a PP2C from rice (*Oryza sativa*), shows strong tolerance to high salinity during germination [23]. ABA receptors from *PYR/PYL/RCAR* family interact with PP2Cs to regulate abiotic stress responses and plant development [24]. In this study, the gene encoding protein phosphatase 2C 6 (TRINITY_DN52120_c0_g1) was upregulated, while the gene encoding abscisic acid receptor PYL4-like (TRINITY_DN8430_c0_g1) was downregulated in response to salt stress during germination (Figure 5B), showing that these two key regulators involved in the ABA signaling pathway may be important to regulate the salt tolerance during seed germination in *S. liaotungensis*.

BRs play important roles in overcoming inhibition of germination by ABA, and BRs’ biosynthesis and signal transduction could improve plant tolerance to salt stress [25,26]. Under salt stress, exogenous application of 24-epibrassinolide (EBR) contributes to promoting seed germination [27]. BKI1 functions as a negative regulator of the BR receptor, brassinosteroid insensitive 1 (BRI1); however, BKI1 is a positive regulator in the salt stress tolerance mechanism [28]. During germination, the gene encoding BRI1 kinase inhibitor 1-like (TRINITY_DN12138_c0_g1) was shown to be up-regulated in response to salt stress (Figure 5B), suggesting that it may activate BR signaling to enhance the salt tolerance during germination. However, the molecular mechanism needs to be further studied.

### 3.2. Transcription Factors Involved in Seed Germination under Salt Stress

Transcription factor families and hormone signaling synergistically regulate seed dormancy and germination [8]. A total of 316 DEGs mapped to 26 TF families were identified in the SlD vs. SlG_C group (Appendix A). Most DEGs were enriched to MYB, AP2/ERF, bHLH, C2C2, bZIP, WRKY, and NAC families, of which, all WRKY TFs were upregulated in the SlD vs. SlG_C group (Appendix A), suggesting that these TF families play essential roles during seed germination. Studies have reported that TF families, such as MYB, WRKY, NAC, and bZIP, are involved in abiotic and biotic stress responses, and many TFs are related to enhanced tolerance in both model and crop plants [29,30]. In our study, 14 TFs were found to be salinity-responsive and differentially expressed during germination (Figure 5A). ATHB-12, an Arabidopsis homeobox-leucine zipper domain protein (HD-Zip) that regulates sodium exclusion and enhances salt tolerance in yeast, is activated by NaCl and ABA treatment [31]. The gene encoding ATHB-12-like (TRINITY_DN8056_c0_g1) was significantly upregulated under salt stress during the germination stage, whereas the gene was not changed in the SlD vs. SlG_C group (Figure 5A), suggesting that ATHB12 is specifically salt-induced and may regulate salt tolerance through ABA signaling pathway during seed germination in *S. liaotungensis*. Among these genes related to ethylene-responsive transcription factors, TRINITY_DN32173_c0_g1 and TRINITY_DN9523_c0_g1 were increased, while TRINITY_DN23862_c0_g1 and TRINITY_DN8900_c0_g2 were repressed under salinity stress during germination (Figure 5A). These transcription factors may act synergistically with ethylene and other hormones to regulate seed germination under salt stress.

Plant growth and development, such as secondary wall formation, lateral root development, and seed germination, are regulated by NAC transcription factors. NAC transcription factors have been shown to enhance salt tolerance in plants. Transcriptomic analysis revealed that *SlNAC2* (TRINITY_DN7999_c0_g1) expression level was significantly increased under salt stress (Figure 5A). Function analysis also showed that overexpression *SlNAC2* significantly increased the germination rate under salt treatment in *Arabidopsis* (Figure 8). These results suggest that *Sl*NAC2 plays a positive regulator role in response to salt stress during germination in *S. liaotungensis*. Plant hormones are involved in seed germination that is regulated by the NAC transcription factor under salt stress. NTL8, an NAC transcription factor, regulates seed germination via GA-mediated salt signals, and the mutation of *NTL8* enhances salt tolerance during germination in *Arabidopsis* [32]. Under high salinity, NAC transcription factor NTM2 regulates germination by integrating auxin and salt signaling through *IAA30* [33]. Therefore, we speculate that *Sl*NAC2 regulates seed germination under salt stress via certain hormonal signaling pathways.

### 3.3. ROS Regulation during Seed Germination under Salt Stress

Under normal circumstances, plants produce ROS and maintain the dynamic equilibrium in vivo. However, under abiotic stresses, especially salt stress, excessive ROS can cause oxidative stress and cellular damage, thereby inhibiting seed germination [34]. From dry seed to germination stage, ROS production begins, and the antioxidant enzyme system is activated to maintain ROS homeostasis. Therefore, many genes encoding SOD, POD, and APX were upregulated in the SlD vs. SlG_C group (Appendix A). In response to salinity, plant antioxidant systems can scavenge extra ROS and alleviate oxidative damage [35]. Antioxidant enzymes including SOD, CAT, POD, and APX have been shown to play different enzymatic functions in plants. Among them, SOD specifically catalyzes two superoxide radicals (O_2_^−^) to form H_2_O_2_ and O_2_, and then H_2_O_2_ is catalyzed to form H_2_O by POD and CAT [36]. Our physiological results showed that, compared with the control condition, POD and CAT activities were considerably elevated during germination under salt stress (Figure 2B,C). Transcriptome analysis also suggested that many genes encoding antioxidant enzymes were significantly induced by salinity, especially the peroxidase family (Figure 6A). This suggests the vital role of the peroxidase family response to salt stress. Among 12 genes encoding peroxidase, three genes (TRINITY_DN10166_c0_g1, TRINITY_DN5883_c0_g1, and TRINITY_DN6393_c0_g1) were increased, while the rest were repressed in response to salt stress (Figure 6A). Peroxidases, due to their double catalytic functions, can eliminate ROS or oxidize compounds, which are engaged in cell wall stiffening processes, including lignification and suberization, cell wall loosening, or defense mechanisms [37]. Therefore, three up-regulated peroxidase genes could be involved in eliminating excess ROS to alleviate oxidative damage, thus enhancing salt tolerance. Peroxidases may not only directly oxidize target molecules by taking advantage of H_2_O_2_, but may also be involved in ROS production [38]. Peroxidases have the capacity to generate H_2_O_2_, and subsequently OH radicals, via the hydroxylic cycle, meanwhile OH radicals could promote seed germination [39]. In this study, H_2_O_2_ content was increased under salt stress during germination compared with the control condition (Figure 2D), suggesting that excess H_2_O_2_ might be conducive to the production of OH radicals that promoted seed germination. The Class III peroxidase (PRX) mutant *atprx16* exhibits a strong germination acceleration phenotype in *Arabidopsis* [40]. The expression levels of other peroxidases were repressed by salt stress, which indicated their involvement in accelerating seed germination. Moreover, several class III *PRX* genes have reported to regulate tolerance to abiotic stresses through activating the downstream stress-related genes and the ABA signaling pathway [41,42]. Hence, ROS, hormone signaling pathway, and peroxidases synergistically regulate germination under salt stress in *S. liaotungensis*.

### 3.4. Cell Wall Remodeling during Seed Germination under Salt Stress

The plant cell wall not only provides mechanical strength and protection for plant tissue but also functions in plant growth and development [43]. Cell wall remodeling-associated proteins, including cellulases, xyloglucan-modifying and pectin-modifying enzymes, and apoplastic reactive oxygen species (aROS) regulate cell wall extensibility, thereby regulating cell enlargement and expansion [44]. Cell wall loosening, stress relaxation, and cell wall extension triggered by water absorption facilitate cell expansion, whereas seed germination requires embryo cell expansion and the weakening of surrounding endospermic tissue [44]. A majority of genes encoding expansin, cellulases, xyloglucan-modifying and pectin-modifying enzymes, and polygalacturonase were significantly upregulated in the SlD vs. SlG_C group (Appendix A). This indicates that cell wall remodeling mainly occurs in the late germination stage. XTHs and expansins are related to cell wall remodeling. XTHs are considered to accelerate the internal breakage of xyloglucan polymers and integrate the newly generated xyloglucan polysaccharide into the cell wall, thereby retaining the thickness and integrity of the cell wall [45]. Five genes encoding xyloglucan endotransglucosylase/hydrolase protein (TRINITY_DN6994_c0_g1, TRINITY_DN6994_c0_g2, TRINITY_DN4215_c0_g1, TRINITY_DN40370_c0_g1 and TRINITY_DN11967_c0_g1) were repressed under salt stress during the germination stage (Figure 6B), which suggested their involvement in cell wall loosening, thus facilitating seed germination. In *Arabidopsis*, overexpression of *XTH23* increases plant salt tolerance through activating the BR signaling pathway [46]. These findings above show that these *XTHs* may regulate seed germination under salt stress via the hormone signaling pathway. Expansins are considered to be involved in cell wall loosening via cleaving linkages between cellulose microfibrils [47]. Overexpression of *RhEXPA4* gene from rose brings about a higher germination rate under salt stress in *Arabidopsis* [48]. Two genes encoding expansin (TRINITY_DN8701_c0_g1 and TRINITY_DN11950_c0_g1) were upregulated by salinity stress (Figure 6B), implying that these expansins may act as positive regulators during germination under salt stress in *S. liaotungensis*.

Polygalacturonases (PGs) is a kind of the hydrolase that cleaves the links between the polymers that make up the cell wall, altering its extensibility and cell-cell adhesion [49]. Five genes encoding polygalacturonase in *S. liaotungensis* were up-regulated by salinity stress (Figure 6B), which might positively regulate seed germination. Pectinesterase (PE), a cell wall-associated protein, is required for demethylating pectin galacturonyl residues, which allows pectin to be aggregated via calcium chelation, thereby strengthening the cell wall [50]. Fruit softening during ripening occurs when the salt-dependent isoform of PE is silenced in tomato [51]. Hence, the genes encoding pectinesterase (TRINITY_DN6078_c0_g1 and TRINITY_DN4991_c0_g1) were upregulated, whereas pectinesterase inhibitor (TRINITY_DN4831_c0_g1 and TRINITY_DN5680_c0_g1) was downregulated by salinity stress (Figure 6B), suggesting that they may play essential roles in strengthening the cell wall, thus enhancing salt tolerance in *S. liaotungensis*. However, their regulatory roles also might partly inhibit seed germination under salt stress. Moreover, apoplastic peroxidases have been reported that are involved in ROS production, such as apoplastic hydroxyl radicals, whereas hydroxyl radicals also lead to cell wall loosening, thereby promoting germination [52,53]. Therefore, the cell wall remodeling proteins play essential and different roles in cell wall loosening during the germination stage, and these proteins may co-regulate seed germination with ROS and hormone signaling pathways under salt stress in *S. liaotungensis*.

### 3.5. Carbohydrate Metabolic Pathway Involved in Seed Germination under Salt Stress

Seeds resume metabolic activity in the imbibition stage, resulting in the decomposition of macromolecules stored in endosperm, which provide the energy and nutrients for seed germination [54]. Starch stored in the endosperm is the major energy source for seed germination. The significant up-regulation of starch catabolism-related genes indicates embryonic energy supply is indispensable to resist damage caused by salt stress [20]. In the SlD vs. SlG_C group, most of the DEGs related to processes, including the glyoxylate cycle, starch synthesis and catabolic process, sucrose synthesis, and glycolytic cycle, were upregulated (Appendix A). Our physiological data showed that soluble sugar content in SlG_C was significantly decreased compared with SlD, which was similar to the result reported by Liu et al. [55], indicating that this could be related to the vigorous metabolism and energy requirement for life activities during seed germination (Figure 2E). However, it has been reported that the content of soluble sugar in dry seeds is significantly lower than that in germinated seeds [56], which may be due to the dynamical change of soluble sugar content during germination, and a different plant species and growing environment, different sampling method and sampling period would make the results different. Under salt stress, two genes encoding important enzymes involved in starch synthesis, such as phosphoglucomutase (PGM) (TRINITY_DN24222_c0_g1) and ADP-glucose pyrophosphorylase (AGPase) (TRINITY_DN480_c0_g1), were upregulated (Figure 6C), suggesting that starch produced anew in the embryo during germination may be an effective strategy for carbohydrate utilization and energy supply. The levels of important Calvin cycle enzymes, including ribulose-1,5-bisphosphate carboxylase/oxygenase (Rubisco), fructose-1,6-bisphosphate aldolase (FBA), and sedoheptulose-1,7-bisphosphate (SBPase), were significantly increased under salt stress (Figure 6C). These enzymes are mainly used for transient starch synthesis in chloroplast, and synthesized intermediates are transported to the cytosol via the triose-phosphate, which is used for sucrose formation, thereby supplying energy for seed germination [57]. Therefore, our physiological data also showed that soluble sugar content was significantly increased under salt stress during germination compared with control (Figure 2E). Chloroplast FBA and cytosol FBA are two isoforms of FBA, whereas metabolites produced by the chloroplast FBA are used for starch synthesis [58]. Several studies have showed that FBA expression level is elevated under high salinity in *Sesuvium portulacastrum*, and overproduced chloroplast FBA could improve resistance to salt stress in tobacco [59,60]. The expression of the gene encoding the chloroplast FBA (TRINITY_DN17700_c0_g1) was increased by salinity stress, which might enhance salt tolerance during seed germination (Figure 6C). In the Calvin cycle’s regenerative phase, the SBPase maintains Rubisco activity by regenerating ribulose-1,5-bisphosphate (RUBP) and preventing Rubisco activase from sequestration to the thylakoid membrane, thereby enhancing photosynthesis’ tolerance to salt stress [61]. In transgenic rice, overexpression of SBPase enhances salt tolerance [61]. These genes encoding SBPase (TRINITY_DN7313_c0_g1) and Rubisco (TRINITY_DN4032_c0_g1) were upregulated by salinity stress (Figure 6C), suggesting that these enzymes may play vital roles in coping with salt stress, thereby promoting seed germination.

The carbohydrate composition consists of oligosaccharides and polysaccharides in seeds. Besides starch, cell wall polysaccharides including cellulose, hemicelluloses, and pectin are present in plant seeds [62]. In the SlD vs. SlG_C group, the expression levels of genes encoding glycosidase and endoglucanase that are involved in polysaccharides hydrolysis were increased (Appendix A). Endoglucanase from the bacterial strain *Martelella endophytica* is shown to be a member of the glycosidic hydrolase family 5 (GH 5) with a catalytic domain, and these enzymes are salt-tolerant at up to 15% NaCl [63]. In this study, endoglucanase (TRINITY_DN5891_c0_g1 and TRINITY_DN4567_c0_g1) was induced by salt stress during germination (Figure 6C), implying that increased endoglucanase expression levels may enhance salt tolerance during germination. The degradation of endosperm cell wall polysaccharides, primarily ß-glucans, is beneficial for seed germination, and these products of cell wall degradation may also provide a significant source of carbohydrates for seed germination. Many enzymes, such as endo-(1-3)-ß-glucanase and ß-glucosidase, complete the hydrolysis of ß-glucans [64]. Many genes encoding endo-(1-3)-ß-glucanase and ß-glucosidase were induced by salinity stress (Figure 6C), suggesting that these proteins may promote seed germination through degrading the endosperm cell wall. Since these enzymes belong to different glycosidic hydrolase families, their functions in this process are different, which needs to be further explored. In addition, the expression of ß-glucosidase is dependent on exogenous GA in germinating seeds of GA-deficient *ga1* mutant [65]. Hence, these enzymes may regulate seed germination through hormone signaling pathways under salt stress. Arabidopsis *hot2* mutant that encodes an endochitinase-like protein shows an aberrant tolerance to salt stress and accumulates high levels of Na^+^ in cells under normal or NaCl stress conditions [66]. In this study, two genes encoding endochitinase (TRINITY_DN6856_c0_g1 and TRINITY_DN3265_c0_g2) were downregulated under salt stress (Figure 6C), suggesting that these proteins may prevent the excessive accumulation of Na^+^ ions in cells, thereby enhancing the salt tolerance.

These results above demonstrate that carbohydrate metabolism is necessary for the energy supply for germination and may combine with hormones and other biological pathways to synergistically regulate seed germination under salt stress in *S. liaotungensis*.

## 4. Materials and Methods

### 4.1. Plant Materials and Growth Conditions

*Suaeda liaotungensis* K. seeds were harvested from coastal saline-alkali soil in Yingchengzi Town, Dalian City, Liaoning Province, China at 121.36°/38.99° (longitude/latitude). All plant samples for this research were obtained from the public land in November 2020. *Suaeda liaotungensis* seeds were collected in that region without regard for endangered flora. The collected seeds were stored in a 4 °C refrigerator for subsequent experiments.

Wild-type controls were Arabidopsis (*Arabidopsis thaliana*) ecotype Columbia-0. *SlNAC2* overexpression transgenic lines (*SlNAC2-OX*) and pBI121 vector plants (VT1 and VT2) in *Arabidopsis* has been reported [15], and T3 homozygous seeds from these transgenic plants were stored in our laboratory. Seeds from these plants above were sterilized in 70% (*v*/*v*) ethanol for 5 min and then in 1% (*v*/*v*) NaClO for 10 min before being rinsed five times with sterile water. These seeds were sown on MS medium containing 1% (*w*/*v*) sucrose and solidified by 0.8% (*w*/*v*) agar after incubation for 2 days at 4 °C. Plates were placed to 22 °C for germination in the incubator. Seedlings were transplanted to soil at 7 days old and cultivated at 22 °C with a 16-h light/8-h dark cycle in the culture chamber.

### 4.2. Seed Germination Assay

For the *S. liaotungensis* brown seed germination test, 35 robust brown seeds were chosen and equally dispersed in the Petri dish, which was covered with two layers of filter paper soaked in 8 mL NaCl solution (200, 400, 500, 600, 800, and 1000 mM) as the treatment and sterile water as the control. Germination was tested for 10 d at a temperature of 10:20 °C with a 12-h light/12-h dark cycle. When testing germination, the weight of Petri dishes (containing filter paper and seeds) was monitored every day, and extra distilled water was applied to manage the moisture of the filter paper. When radical protrusion was visible, seeds were deemed germinated. The percentage of seeds that germinated after 10 d was calculated. For the germination experiment, four biological replicates were carried out.

For the *A.*
*thaliana* seed germination test, seeds from wild-type, pBI121 plants (VT1 and VT2), and *SlNAC2-OX* transgenic plants were surfaced sterilized and sowed on MS solid medium with or without 150 mM NaCl. The plates were placed in the incubator for 7 d to germinate before being photographed. When radical protrusion was visible in microscope, seeds were considered germinated. For the wild-type and transgenic *A. thaliana* lines, three biological replicates were performed. Each line contained 60 seeds.

### 4.3. Imbibition Tests

In *S. liaotungensis*, 30 brown seeds from each group were used in an imbibition test. The dry mass of brown seeds was measured (time 0), and then seeds were evenly spread in the Petri dish using two layers of filter paper soaked in distilled water or 500 mM NaCl solution. The seeds were taken from the Petri plates after 1 h, wiped dry using filter paper, reweighed, and placed back in the Petri dishes. After 2, 4, 6, 8, 12, 24, and 48 h of water absorption, the seed mass was measured again. The seeds had started to germinate intensively with distilled water and 500 mM NaCl at 24 h and 48 h, respectively. Relative increase in seed fresh weight was calculated according to Baskin’s method [67]. Three biological replicates were performed in this experiment.

### 4.4. Determination of Physiological Indexes

The antioxidant activity determination method was slightly modified according to Wang’s method [68]. Approximately 0.2 g seeds for each sample were homogenized in a pre-cooled phosphate buffer solution (PBS) and centrifuged at 4 °C for 30 min (12,000 rpm/min for SOD, 6000 rpm/min for POD and CAT), and the supernatant was designated as the crude extract of antioxidant enzymes.

For the measurement of SOD activity, 3 mL reaction solution was mixed in 0.05 M PBS (pH 7.8) containing 130 mM methionine, 750 μM nitrotetrazolium blue (NBT), 100 μM EDTA-Na_2_, 20 μM riboflavin and 50 μL enzyme extract, and the absorbance was detected at 560 nm. For the measurement of CAT activity, the 3 mL reaction solution including 0.05 M PBS (pH 7), 0.3% (*v*/*v*) H_2_O_2_, and 100 μL enzyme extract was prepared, and then the decline of H_2_O_2_ absorbance at 240 nm was detected. For the determination of POD activity, the 3 mL reaction solution including 0.05 M PBS (pH 5.5), 2% (*v*/*v*) H_2_O_2_, 0.05 M 2-methoxyphenol, and 100 μL enzyme extract was prepared, and then the change in absorbance value at 470 nm was detected.

H_2_O_2_ content was conducted using the H_2_O_2_ Content Test Kit (Solarbio, Beijing, China). The content of H_2_O_2_ was determined by spectrophotometry. H_2_O_2_ and titanium sulfate produce yellow peroxide complex with characteristic absorption at 415 nm. Seeds were homogenized by adding 1 mL acetone to an ice bath. The homogenate was transferred into the centrifuge tube and centrifuged at 8000× *g* at 4 °C for 10 min. Subsequently, two reagents in the kit according to the instructions were added to the supernatant, and centrifuged at 4000× *g* at room temperature for 10 min. The precipitate was dissolved with the reagent in the kit and stood at room temperature for 5 min. Finally, the absorbance values were measured at 415 nm wavelengths using the spectrophotometer.

Soluble sugar content was measured by the method described previously [55,56,69]. Proline content was determined by the method described previously [70].

### 4.5. Sample Collection for the Transcriptome Sequencing Analysis

The RNA-seq experiment was carried out on dry and germinating brown seeds of *S. liaotungensis* under distilled water and salt solution treatment (500 mM NaCl), namely the DS stage (dry seed stage) and GS stage (radicle protrusion about 5 mm in concentrated germination stage). The dry brown seeds of *S. liaotungensis* were labeled as SlD, and the germination seeds under the distilled water and 500 mM NaCl treatment were labeled as SlG_C and SlG_N, respectively. Each sample contained three biological replicates, each of which with approximately weighed 0.2 g seeds. For transcriptome sequencing and determination of physiological experiments, these materials were frozen in liquid nitrogen and stored at −80 °C.

### 4.6. Library Construction, Sequencing and Transcriptome De Novo Assembly

Total RNA for Illumina sequencing was extracted from the seeds of SlD, SlG_C, and SlG_N samples. The quantity and quality of total RNA were detected using the Nanodrop 2000 spectrophotometer (Thermo Scientific, Waltham, MA, USA). Each sample’s mRNA was extracted from total RNA, which was then fragmented into tiny fragments (300 bp). Under the action of reverse transcriptase, the random hexamers were added to synthesize the cDNAs using mRNA as the template. End-repaired 3′ adenylated cDNAs were generated and ligated using sequencing adaptors. PCR was used to enrich the isolated cDNA fragments. Then, these cDNA libraries were sequenced using Illumina Novaseq 6000 sequencer. The image data from the sequencer was converted into sequence data (raw reads) using CASAVA base calling and saved in fastq format. The raw paired end reads were trimmed and quality controlled by fastp (https://github.com/OpenGene/fastp, 5 April 2021) with default parameters. Then clean data from the samples were used to do de novo assembly with Trinity (http://trinityrnaseq.sourceforge.net/, 5 April 2021) [71]. To create contigs, the Trinity program first merged clean reads with their overlapping regions. De Bruijn graphs were created as a result of grouping these contigs into components. Finally, neither end of the sequences could be extended, and then these sequences were named as unigenes. The assembled unigenes were assessed and optimized with BUSCO (Benchmarking Universal Single-Copy Orthologs, http://busco.ezlab.org, 5 April 2021), TransRate (http://hibberdlab.com/transrate/, 5 April 2021), and CD-HIT (http://weizhongli-lab.org/cd-hit/, 5 April 2021). All assembled unigenes were searched against six databases including the Clusters of NCBI non-redundant protein (NR), Gene Ontology (GO), Kyoto Encyclopedia of Genes and Genomes (KEGG), Orthologous Groups (COG), Swiss-Prot, and Protein family (Pfam) databases using BLASTX to identify the proteins that had the highest sequence similarity with the given unigenes to retrieve their function annotations, and a typical cut-off E-values less than 1.0 × 10^−5^ was set.

### 4.7. Analysis of Differentially Expression Genes (DEGs) and Functional Enrichment

To identify the DEGs among two contrasting groups (SlD vs. SlG_C and SlG_C vs. SlG_G, three biological replicates per group), the TPM (Transcript Per Million reads) method was used to analyze the expression level. RSEM (http://deweylab.biostat.wisc.edu/rsem/, 5 April 2021) was used to quantify gene abundances. Paired end reads, length of reads, length distribution of fragments, and quality values were considered to distinguish transcripts of different isoforms of the same gene. DESeq2 software was used to statistically evaluate raw counts using the negative binomial distribution. The genes with *P*-adjust < 0.05 and |log_2_fold-change| ≥ 1 were regarded DEGs. Then, these DEGs were analyzed for GO and KEGG enrichment using the Goatools software by a corrected *P*-adjust < 0.05.

### 4.8. RT-PCR and Reverse Transcription Quantitative PCR (RT-qPCR)

To validate the *SlNAC2* transcript level in wild-type, pBI121 plants, and overexpression transgenic lines of *SlNAC2*, the Plant Total RNA Purification Kit (Tiangen, Beijing, China) was used to extract total RNA from 7-day-old seedlings. Then, using the reverse transcription kit (TransGen Biotech, Beijing, China), 1.5 ug RNA was reverse-transcribed into cDNA. The *AtACTIN2* (NM_112764) was used as an internal reference gene. RT-PCR primers are presented in Appendix A.

To validate the DEGs using RT-qPCR, 15 genes responding to salt stress were selected from the RNA-seq studies. Total RNA samples from SlD, SlG_C, and SlG_N were extracted as stated in the Plant Total RNA Purification Kit (Tiangen, Beijing, China). An amount of 1.5 ug RNA was converted into cDNA using the reverse transcription kit (TransGen Biotech, Beijing, China). RT-qPCR experiments were conducted using fluorescence quantitative PCR kit (TransGen Biotech, Beijing, China), and then analyzed via the Real-Time PCR machine (Takara, Japan). The cycle threshold (Ct) 2^−ΔΔCt^ method was used to calculate the relative expression levels of these genes. The *SlACTIN* (GenBank No. JX860282) was used as an internal reference gene. The primers for RT-qPCR are presented in Appendix A.

### 4.9. Statistical Analysis

Statistical analysis was performed by ANOVA using the SPSS 25.0 software, and *p* < 0.05 was considered significant. All data were presented as means ± SD from three independent biological replicates.

## 5. Conclusions

Seed germination is closely related to internal molecular mechanisms and external environmental factors. This is the first report on the molecular mechanisms of seed germination in response to salt stress in *S. liaotungensis* in terms of the transcriptomic level, combined with physiological and functional verification analysis. Many genes and pathways associated with salt response were detected. The complicated mechanisms of salt tolerance were deployed via some candidate genes involved in plant hormone and transcription regulation, ROS metabolism, cell wall organization or biogenesis, and carbohydrate metabolism during seed germination (Figure 9). Moreover, overexpression *SlNAC2* enhanced salt tolerance during germination stage in *Arabidopsis*. Therefore, this research not only deepens our understanding of the mechanism of salt tolerance during seed germination, but also provides important information to excavate salt tolerance genes in *Suaeda liaotungensis*.

## Figures and Tables

**Figure 1 ijms-23-12229-f001:**
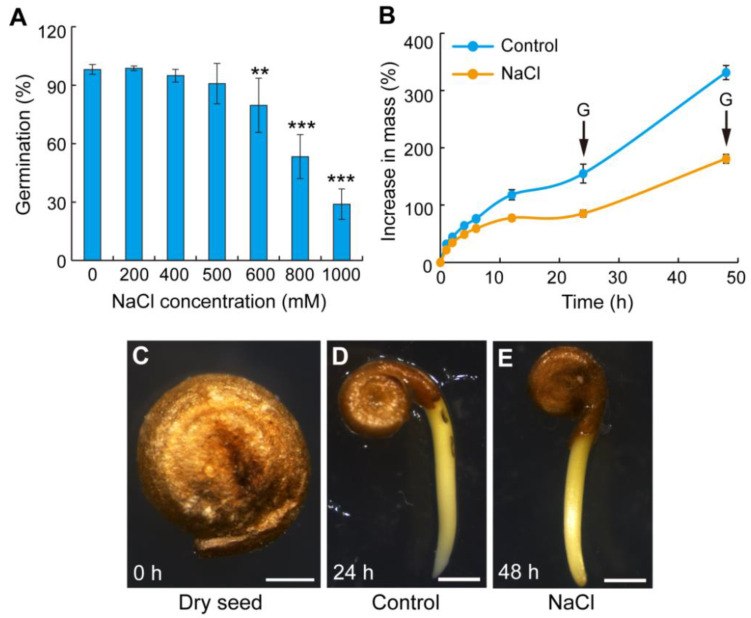
Identification of salinity concentration and seed morphology during brown seed germination in *S. liaotungensis*. (**A**) Seed germination rate under the control condition (distilled water) and different NaCl concentration treatment. Seed germination rates were calculated for 10 d. Values presented are the means ± SD for four independent experiments (*n* = 40 seeds for each treatment). Significant differences were analyzed using one-way ANOVA followed by LSD test (** *p* < 0.01; *** *p* < 0.001). (**B**) Imbibition curves for brown seeds of *S. liaotungensis* under the control condition and 500 mM NaCl treatment. Values presented are the means ± SD for four independent experiments (*n* = 30 seeds for each treatment). “G”, concentrated germination time. (**C**–**E**) Phenotypes of dry seeds, germinated seeds under the control (24 h) and 500 mM NaCl (48 h) treatment. Bars = 0.5 mm in (**C**) and 1 mm in (**D**,**E**).

**Figure 2 ijms-23-12229-f002:**
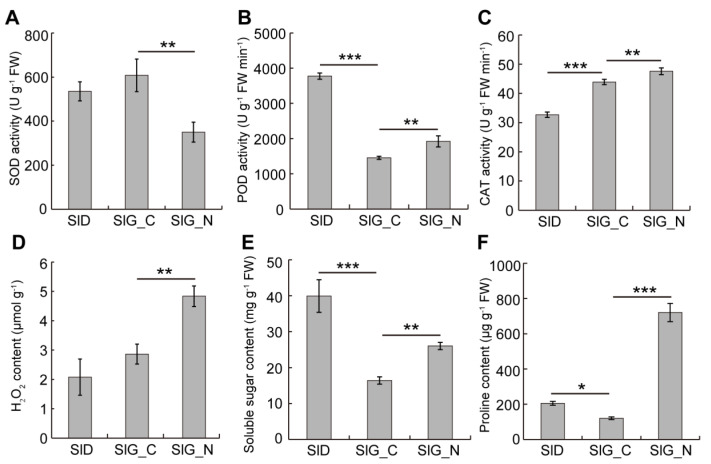
Determination of physiological indexes in SlD, SlG_C, and SlG_N. (**A**) SOD activity. (**B**) POD activity. (**C**) CAT activity. (**D**) H_2_O_2_ content. (**E**) Soluble sugar content. (**F**) Proline content. Values presented are the means ± SD for three independent experiments. Significant differences were analyzed using one-way ANOVA followed by LSD test (* *p* < 0.05; ** *p* < 0.01; *** *p* < 0.001).

**Figure 3 ijms-23-12229-f003:**
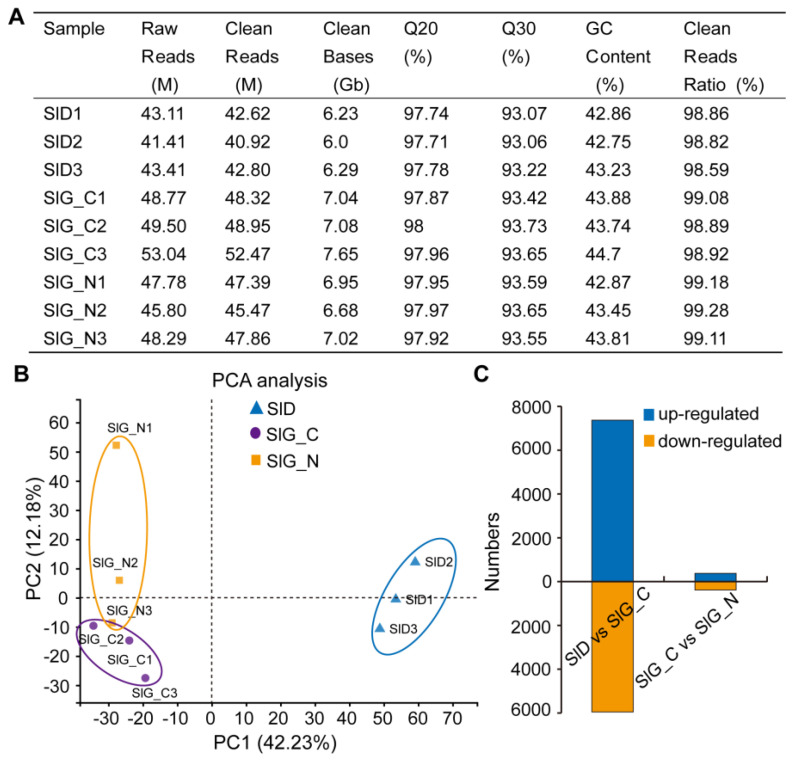
Summary statistics of the transcriptome of SlD, SlG_C, and SlG_N in *S. liaotungensis*. (**A**) Transcriptome assembly and annotated statistics. GC is short for guanine-cytosine. (**B**) PCA (principal component analysis) of three duplicate samples from SlD (blue), SlG_C (purple), and SlG_N (orange) group. (**C**) The numbers of up-regulated (blue) and down-regulated (orange) genes in the SlD vs. SlG_C and SlG_C vs. SlG_N groups.

**Figure 4 ijms-23-12229-f004:**
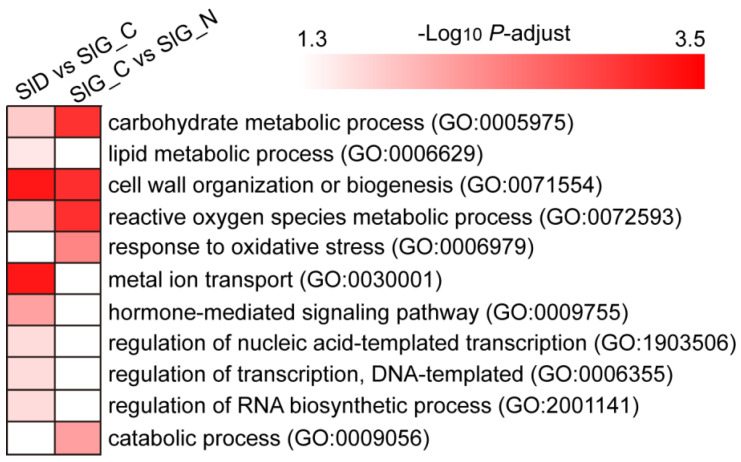
GO enrichment analysis of the DEGs identified in the SlD vs. SlG_C and SlG_C vs. SlG_N groups. Colors of boxes indicate the *P*-adjust of enrichment.

**Figure 5 ijms-23-12229-f005:**
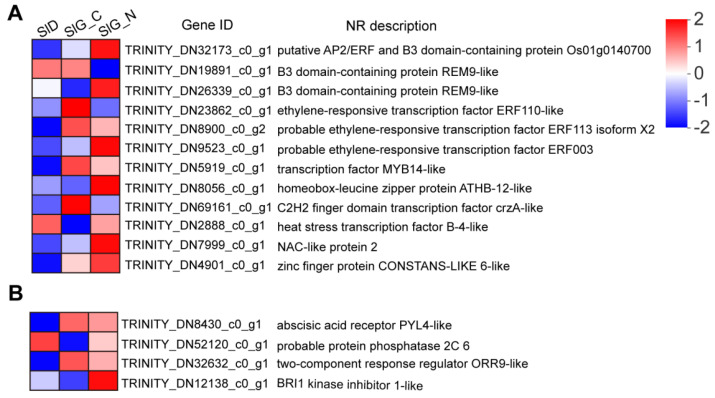
Heatmap of differential expression genes related to the transcriptional regulation (**A**) and hormone signal pathway (**B**) in SlD, SlG_C, and SlG_N. The values of the blue to red gradient bar indicate the log_10_ fold change in expression.

**Figure 6 ijms-23-12229-f006:**
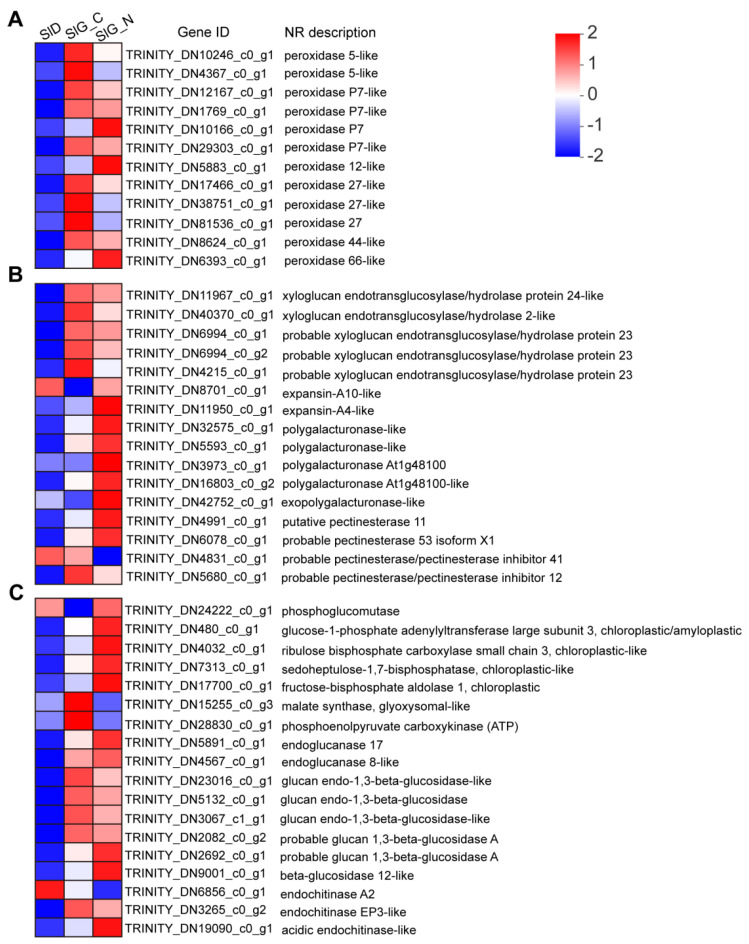
Heatmap of differential expression genes related to the reactive oxygen species metabolic process (**A**), cell wall organization or biogenesis (**B**), and carbohydrate metabolic process (**C**) in SlD, SlG_C, and SlG_N. The values of the blue to red gradient bar indicate the log_10_ fold change in expression.

**Figure 7 ijms-23-12229-f007:**
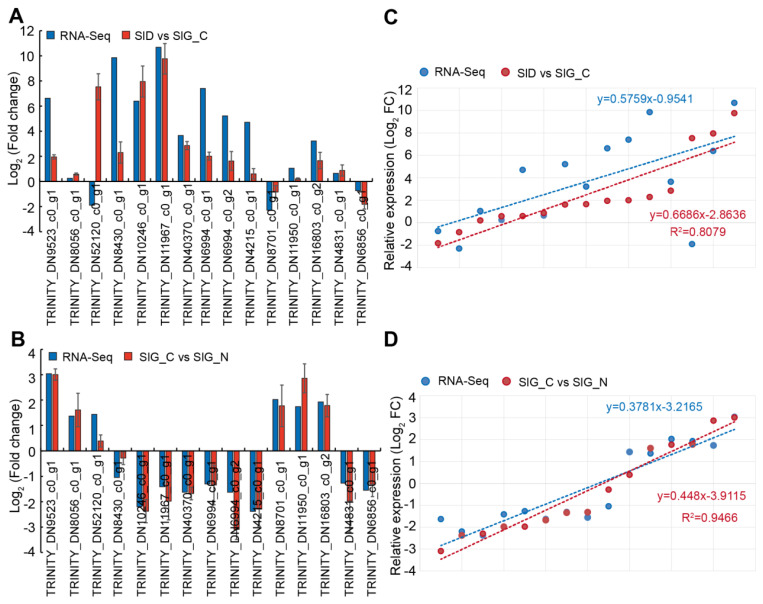
RT–qPCR validation of 15 candidate DEGs in the SlD vs. SlG_C (**A**) and SlG_C vs. SlG_N (**B**) groups. Correlation between RNA–seq and RT–qPCR in the SlD vs. SlG_C (**C**) and SlG_C vs. SlG_N (**D**) groups. These 15 genes were: TRINITY_DN9523_c0_g1, probable ethylene-responsive transcription factor ERF003; TRINITY_DN8056_c0_g1, homeobox-leucine zipper protein ATHB-12-like; TRINITY_DN52120_c0_g1, TRINITY_DN8430_c0_g1, abscisic acid receptor PYL4-like; TRINITY_DN10246_c0_g1, peroxidase 5-like; TRINITY_DN11967_c0_g1, xyloglucan endotransglucosylase/hydrolase protein 24-like; TRINITY_DN40370_c0_g1, xyloglucan endo–transglucosylase/hydrolase 2-like; TRINITY_DN6994_c0_g1, TRINITY_DN6994_c0_g2, and TRINITY_DN4215_c0_g1, probable xyloglucan endotransglucosylase/hydrolase protein 23; TRINITY_DN8701_c0_g1, expansin-A10-like; TRINITY_DN11950_c0_g1, expansin-A4-like; TRIN–ITY_DN16803_c0_g2, polygalacturonase At1g48100-like; TRINITY_DN4831_c0_g1, probable pec–tinesterase/pectinesterase inhibitor 41 and TRINITY_DN6856_c0_g1, endochitinase A2. The ex–pression of these genes was normalized by *SlACTIN* gene.

**Figure 8 ijms-23-12229-f008:**
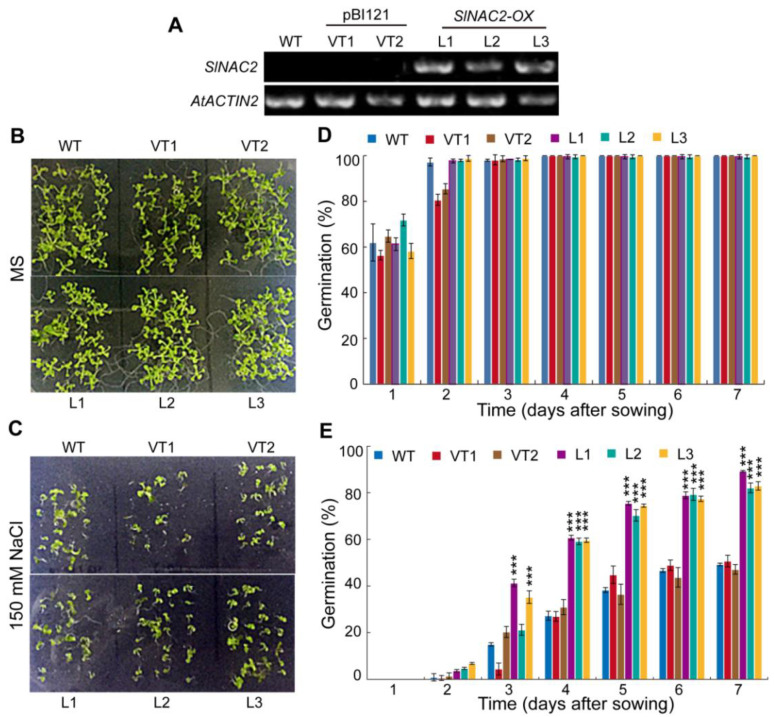
Functional characterization of *Sl*NAC2 in seed germination under salinity stress in *Arabidopsis*. (**A**) RT-PCR analysis of the *SlNAC2* expression level from wild-type (Col-0), pBI121 plants (VT1 and VT2), and *SlNAC2* transgenic plants (*SlNAC2-OX*) in *Arabidopsis*. *AtACTIN2* as the internal control. (**B**,**C**) Phenotypes of seed germination at 7 DAS from wild-type (Col-0), pBI121 plants (VT1 and VT2), and *SlNAC2-OX* plants in MS medium with or without 150 mM NaCl treatment. (**D**,**E**) Germination rates were calculated from wild-type (Col-0), pBI121 plants (VT1 and VT2), and *SlNAC2-OX* plants in MS medium with or without 150 mM NaCl treatment. Germinated seeds were scored daily, and radicle protrusion was used as the morphological marker. Values presented are the means ± SD for three independent experiments (*n* = 60 seeds for each line). Significant differences were analyzed using one-way ANOVA followed by LSD test (*** *p* < 0.001). *OX*, overexpression.

**Figure 9 ijms-23-12229-f009:**
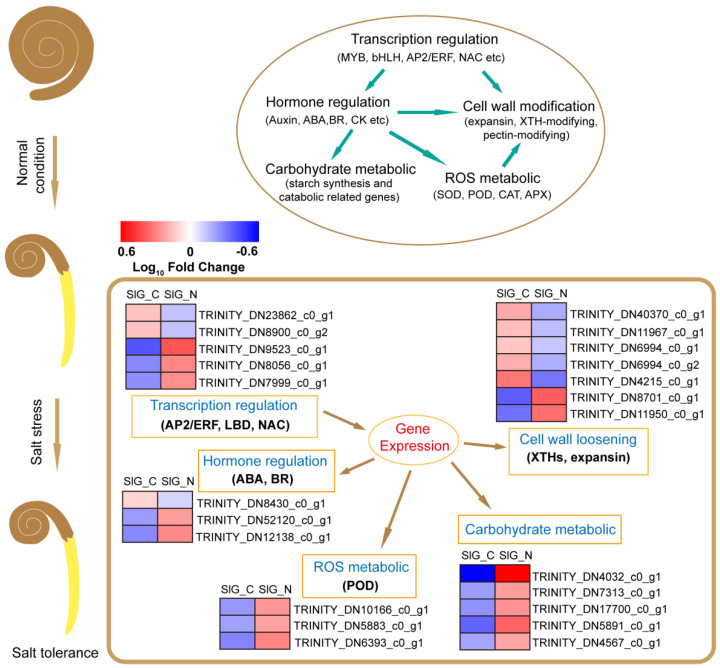
Schematic representation of the molecular response and mechanisms underlying tolerance to salt stress during brown seed germination in *S. liaotungensis*.

## Data Availability

The clean data presented in this study are available in the Sequence Read Archive (SRA) database in NCBI (Accession No. PRJNA873907).

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
