# Peer review of "Transcriptome Analysis Reveals Genes and Pathways Associated with Salt Tolerance during Seed Germination in Suaeda liaotungensis"

_ijms, 2022, doi:10.3390/ijms232012229_

Round 1

Reviewer 1 Report

This study describes seed germination in Suaeda liaotungensis (Kitag) a succulent halophyte of the Chenopodiaceae family. Seeds collected in the wild were germinated with increasing salt (NaCl) concentration. The authors show a remarkable tolerance to salt and precisely analyse the germination process under high salt concentration (500 mM). They perform a quantitative analysis of oxidative stress through antioxidant enzyme activity measurements. They further analyse the process trough transcriptome analyses of the germination process (dry seeds vs germinated seeds comparison) and of the salt stress (germinated seeds under control conditions (water) vs germinating seeds in 500mM NaCl). As Kitag is not a model species, the authors had to go through the difficult task of de novo transcriptome assembly from pooled data. They generated a reference unigene set which was used as a reference set for differential gene expression. They identified 13314 DEG during Kitag germination process and 755 DEG during germination under salt stress.

Although this is valuable and interesting data, I have major concerns regarding how it is analysed. The experimental procedures and the bioinformatics analyses both lack precision in their description. Furthermore, the reference sets used for partial putative transcript gene annotation are very generic and it is very difficult thus to assess the overall dataset quality.

I think this study would be much better with a more thorough analysis relying on existing datasets in the family, such as the Beta genome or salt stress datasets already available on Suaeda genus.

Introduction :

The introduction would benefit from more background details on Suaeda. What is the ploidy level? Is it highly heterozygous, as suggested by the phenotypic diversity of wild collected seeds? What is the number of chromosomes?

Results:

Physiology : the paragraphs on seed germination and the physiology of resistance to salt stress are sound and clear.

Transcriptome analyses: this part has to be analysed more in details. Transcript contigs (“unigenes”) were built using the Trinity software. Wild species are generally a challenge for reference transcriptome assembly due to heterozygosity and unavailability of double-haploid lines (this is why it would be nice to know the ploidy and the level of heterozygosity in the background).

Number of unigenes is high: this might render read quantification very biased. What is the rule for closely related contigs with highly similar sequences, how do you count the reads attributed to one contig? This should be clarified in the text.

In the absence of a specific reference genome for comparison, the authors should also analyse their data against the transcripts of the closely related species: beta, for example, has an assembled genome. Gene models are complete and the study would benefit from comparing.

Furthermore, salt stress data on sugar beet roots , on spinach, is publicly available, as well as salt stress data on other Suaeda species.It would be interesting to check if similar pathways involving orthologs of published putative salt-stress induced genes in this plant family are also involved here.

Reviewer 2 Report

Review of manuscript entitled “Transcriptome analysis reveals genes and pathways associated with salt tolerance during brown seed germination in Suaeda liaotungensis”

 In this work authors aim to elucidate mechanisms underlying salt stress tolerance of halophyte Suaeda liaotungensis during seed germination state. For this prupoose they utilize transcriptomic analysis with RNA-seq and some biochemical analysis regarding ROS, antioxidant enzyme activity and sugar contents.

Using halophytes to understand plant response to salinity is a powerful tool and has the potential to reveal new genes or functions for genes and tolerance mechanisms. Suaeda liaotungensis is a widely studied halophyte with various publications in the literature. However, its response during germination has not been studied at transcriptomic level. Brown and black seeds with different salt stress tolerance is rather interesting and it would be great to see comparison of two seed types instead of dry vs imbibed seeds (please see below for an important technical question related to this). Anyway, transcriptomic data presented in the study provides insights into tolerance mechanisms at the germination state. Moreover, authors measure activities of antioxidant enzymes, although it is just a glimpse of the whole antioxidant machinery and not sufficient to draw conclusions it demonstrates a differential response between saline and non-saline conditions. As for the portion related to NAC transcription factor (NAC2), in my opinion it does not provide too much novelty to the manuscript since role of SlNAC1 (Li et al. 2014), SlNAC2 (Yang et al. 2014), SlNAC7 (Wang et al. 2021), SlNAC8 (Wu et al. 2018) and SlNAC10 (Du et al. 2022) has been demonstrated previously. Authors indicate that role of NAC2 during germination has not been studied before, which is correct. However, they already demonstrated that NAC7 and NAC8 improve germination rates in Arabidopsis. Hence, I do not think that part related to NAC2 over-expression in Arabidopsis contributes much to the field.

Overall, I think manuscript deserves publication since it provides novel data related to germination stage. However, there are still points that needs to be addressed (listed below).

 -Title, I think “brown” can be removed from the manuscript title since very limited number of people will have prior knowledge of dimorphic seeds of Suaeda and their different salt tolerance.

-L27, “increasingly agriculture”, English

-L30, “seawater containing small amounts of sodium chloride”, actually seawater contain a high concentration usually around 400-600 mM NaCl.

-L33, “50% of cultivated land would be gone”, seems rather informal please revise.

-L35, “to improve the problem of soil salinization” salt tolerant crops would increase yield but not change soil salinization which is caused by other factors.

-L52, “substance”, maybe a term like “resources” is a better fit here.

-As for the dry vs seeds in distilled water. Authors indicate that they use 0.2 g of seeds for isolations. Since seeds are dry it means that there is about 9-10 fold more material per 0.2 g seeds in dry seeds when compared to imbibed seeds. This might be partly normalized during library preparation or during bioinformatic analysis. However, in dry seeds the tissue at hand is only embryo, while in germinated seeds the tissue at hand is quite different due to radicle growth and utilization of the endosperm. Correct comparison in this case would be imbibed but not germinated seeds instead of seeds with radicle protrusion. Authors should state why they use the current strategy and how do they relate their findings in this sense.

This point is also true for enzyme activity and quantitative measurements related to sugars and proline. If authors use 0.2 g seeds then inevitably they used about 9-10 times more material for dry seeds and since they represent their data in ug per g weight it would be misleading. Please also clarify this point.

-Quality of photos in Fig 8 does not let the reader to see germination state of the seeds clearly.

-Please provide more information regarding method used for H2O2 measurements, at least principle used in the kit should be mentioned.

Reviewer 3 Report

1. Title: It needs to point out that the function of this gene was evaluated using the system of Arabidopsis but not in Suaeda liaotungensis.

2. Avoid using first-person writing throughout the manuscript.

3. The authors need to describe the results, abstract, and conclusions according to statistics using "significant" higher, lower, etc. or non-significant" but not "obvious advantages", etc.

4. Figure 2A: There is a lack of significant difference between means.

5. Figur 7D & E: Most of the data in this study look to have small standard errors, however, in figure 7D & E, it looks like the standard errors are large, why?

6. M&M: The part of the transformation event should be described in more detail, including the genetic construct, and validation of the success of transformation using DNA, RNA, and protein-level experiments.

7. Why the authors did not test the gene function using Suaeda liaotungensis as plant material?

8. Figure 10: I don't understand the logic of "technology roadmap", the authors need to rewrite the figure legend. In my opinion, this is a conclusive graph for the possible mechanism related to tolerance of drought and salt stresses in Arabidopsis.

9. Figure 8: Gene names should be italic and also check throughout the manuscript.

10. Page 12: "In summary, our study shows that SlNAC10 can improve the tolerance of transgenic Arabidopsis to salt and drought stresses. Therefore, we believe that SlNAC10 has a potential utility as a natural novel resource for the cultivation of new crop varieties in saline or arid regions." This sentence is redundant and looks similar to the part of the conclusion.

Round 2

Reviewer 1 Report

The authors have improved the paper with more detailed background matter in the intruduction and more precisions in the methods part. this work is providing a valuable dataset on a newly explored species. The manuscript is getting better, however my feeling is that it still lacks a final twist, maybe using a comparison with existing salt stress datasets in the chenopodiaceae family.

Reviewer 2 Report

Authors address majority of points previously raised. However, revised version does not address two remaining major issues given below (Point 7 and Point 8). Moreover, I do not think that responses given by the authors is satisfactory to resolve these issues. Authors should discuss or mention these points in their manuscript otherwise results would be misleading for the reader.

Point 7: -As for the dry vs seeds in distilled water. Authors indicate that they use 0.2 g of seeds for isolations. Since seeds are dry it means that there is about 9-10 fold more material per 0.2 g seeds in dry seeds when compared to imbibed seeds. This might be partly normalized during library preparation or during bioinformatic analysis. However, in dry seeds the tissue at hand is only embryo, while in germinated seeds the tissue at hand is quite different due to radicle growth and utilization of the endosperm. Correct comparison in this case would be imbibed but not germinated seeds instead of seeds with radicle protrusion. Authors should state why they use the current strategy and how do they relate their findings in this sense.

Response 7: In dry seeds the tissue at hand is only embryo, while in germinated seeds the tissue at hand is quite different due to radicle growth and utilization of the endosperm. Actually, we mainly wondered that how changed in the transcriptome of germinating seeds. Therefore, transcriptome comparison between dry seeds and germinated seeds in distilled water were conducted to find out which genes were changed during germination and which important pathways were involved in the germination process. It is very important to reveal the salt tolerance mechanism of seed germination later. If the transcriptome data of imbibition seeds were added, the salt tolerance mechanism of Suaeda liaotungensis seeds during germination could be more fully revealed.

Point 8: This point is also true for enzyme activity and quantitative measurements related to sugars and proline. If authors use 0.2 g seeds then inevitably they used about 9-10 times more material for dry seeds and since they represent their data in ug per g weight it would be misleading. Please also clarify this point.

Response 8: For dry seeds, 0.2 g seeds were used about 9-10 times more material. According to the methods in Song and Bates’ literature, the weight of samples involved in the calculation of sugar and proline content, so the initial weight of two groups of samples need to be unified for subsequent determination

Reviewer 3 Report

The authors did not give a response to my comments.

Round 3

Reviewer 3 Report

It has been improved.